# TRANSFERABLE RECOGNITION-AWARE IMAGE PROCESSING

## ABSTRACT

Recent progress in image recognition has stimulated the deployment of vision systems at an unprecedented scale. As a result, visual data are now often consumed not only by humans but also by machines. Existing image processing methods only optimize for better human perception, yet the resulting images may not be accurately recognized by machines. This can be undesirable, e.g., the images can be improperly handled by search engines or recommendation systems. In this work, we propose simple approaches to improve machine interpretability of processed images: optimizing the recognition loss directly on the image processing network or through an intermediate transforming model. Interestingly, the processing model's ability to enhance recognition quality can *transfer* when evaluated on models of different architectures, recognized categories, tasks and training datasets. This makes the solutions applicable even when we do not have the knowledge of future recognition models, e.g., if we upload processed images to the Internet. We conduct experiments on multiple image processing tasks, with ImageNet classification and PASCAL VOC detection as recognition tasks. With our simple methods, substantial accuracy gain can be achieved with strong transferability and minimal image quality loss. Through a user study we further show that the accuracy gain can transfer to a black-box, third-party cloud model. Finally, we try to explain this transferability phenomenon by demonstrating the similarities of different models' decision boundaries. Code is available at https://github.com/anonymous20202020/Transferable_RA.

## 1 INTRODUCTION

Unlike in image recognition where a neural network maps an image to a semantic label, a neural network used for image processing maps an input image to an output image with some desired properties. Examples include super-resolution (Dong et al., 2014), denoising (Xie et al., 2012), deblurring (Eigen et al., 2013), colorization (Zhang et al., 2016).

The goal of such systems is to produce images of high perceptual quality to a human observer. For example, in image denoising, we aim to remove noise that is not useful to an observer and restore the image to its original "clean" form. Metrics like PSNR/SSIM (Wang et al., 2004) are often used (Dong et al., 2014; Tong et al., 2017) to approximate human-perceived similarity between the processed and original images,

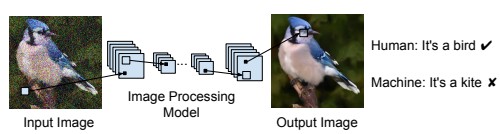

Figure 1: Image processing aims for images that look visually pleasing for human, but not those accurately recognized by machines. In this work we try to enhance output images' recognition accuracy. Zoom in for details.

and direct human assessment on the fidelity of the output is often considered the "gold-standard" (Ledig et al., 2017; Zhang et al., 2018b). Therefore, techniques have been proposed to make outputs look perceptually pleasing to humans (Johnson et al., 2016; Ledig et al., 2017; Isola et al., 2017).

However, while looking good to humans, image processing outputs may not be accurately recognized by image recognition systems. As shown in Fig. 1, the output image of an denoising model could easily be recognized by a human as a bird, but a recognition model classifies it as a kite. One could separate the image processing task and recognition task by specifically training a recognition model on these output images produced by the denoising model to achieve better performance on such

images, or could leverage domain adaptation approaches to adapt the recognition model to this domain, but the performance on natural images can be harmed. This retraining/adaptation scheme might also be impractical considering the significant overhead induced by catering to various image processing tasks and models.

With the fast-growing size of image data, images are often "viewed" and analyzed more by machines than by humans. Nowadays, any image uploaded to the Internet is likely to be analyzed by certain vision systems. Therefore, it is of great importance for the processed images to be recognizable not only by humans, but also by machines. In other words, recognition systems (e.g., image classifier) should be able to accurately explain the underlying semantic meaning of the image content. In this way, we make them easier to search, recommended to more interested audience, and so on, as these procedures are mostly executed by machines based on their understanding of the images. Therefore, we argue that image processing systems should also aim at machine recognizability. We call this problem "Recognition-Aware Image Processing".

It is also important that the enhanced recognizability is not specific to any concrete recognition model, i.e., the improvement is only achieved when the output images are evaluated on that particular model. Instead, the improvement should ideally be transferable when evaluated on different models, to support its usage without access to possible future recognition systems, for example if we upload it to the Internet or share it on social media. This is another case where we cannot separate the processing and recognition task by training them individually, since the recognition is not in our control. We may not know what network architectures (e.g. ResNet or VGG) will be used for inference, what object categories the model recognizes (e.g. animals or scenes), or even what task will be performed (e.g. classification or detection). Without these specifications, it might be hard to enhance image's machine semantics.

In this work, we introduce simple yet highly effective approaches to make image processing outputs more accurately recognized by downstream recognition systems, transferable among different recognition architectures, categories, tasks and training datasets. The approaches we propose add a recognition loss optimized jointly with the image processing loss. The recognition loss is computed using a fixed recognition model that is pretrained on natural images, and can be done without further supervision from class labels for training images. It can be optimized either directly by the original image processing network, or through an intermediate transforming network. Interestingly, the accuracy gain transfers favorably among different recognition model architectures, object categories, and recognition tasks, which renders our simple solutions effective even when we do not have access to the recognition model. Note that our contribution in these approaches does not lie in designing novel network architectures or training procedures, but in making the processed images more accurately recognized based on existing architectures/procedures. We also view our method's simplicity mainly as a strength, as it is easy to be deployed and could serve as a baseline in this new problem.

We conduct extensive experiments, on multiple image processing (super-resolution, denoising, JPEG-deblocking) and downstream recognition (classification, detection) tasks. The results demonstrate our methods can substantially boost the recognition accuracy (e.g., up to 10%, or 20% relative gain), with minimal loss in image quality. Results are also compared with alternative approaches in Appendix A. We explore the transferability phenomenon in different scenarios (architectures, categories, tasks/datasets, black-box models) and demonstrate models' decision boundary similarities to give an explanation. To our best knowledge, this is the first study on transferability of accuracy gain from the processing models trained with recognition loss. Our contributions can be summarized as:

- We propose to study the problem of enhancing the machine interpretability of image processing outputs, a desired property considering the amount of images analyzed by machines nowadays.

- We propose simple yet effective methods towards this goal, suitable for different use cases. Extensive experiments are conducted on multiple image processing and recognition tasks.

- We show that using our simple approaches, the recognition accuracy improvement could transfer among recognition architectures, categories, tasks and datasets, a desirable behavior making the proposed methods applicable without access to downstream recognition models.

- We provide decision boundary analysis of recognition models and show their similarities to gain a better understanding of the transferability phenomenon.

## 2 RELATED WORK

Image processing/enhancement problems such as super-resolution and denoising have a long history (Tsai, 1984; Park et al., 2003; Rudin et al., 1992; Candès et al., 2006). Since the initial success of deep neural networks on these problems (Dong et al., 2014; Xie et al., 2012; Wang et al., 2016b), a large body of works try to investigate better model architecture design and training techniques (Dong et al., 2016; Kim et al., 2016b; Shi et al., 2016; Kim et al., 2016a; Lai et al., 2017; Chen et al., 2018) These works focus on generating high visual quality images under conventional metrics (PSNR/SSIM) or human evaluation, without considering recognition performance on the output.

There are also a number of works that relate image recognition with image processing. Some works (Zhang et al., 2016; Larsson et al., 2016; Zhang et al., 2018c; Sajjadi et al., 2017; Lee et al., 2019) use image recognition accuracy as an evaluation metric for image colorization/super-resolution/denoising, but without optimizing for it in training. Wang et al. (2016a); VidalMata et al. (2019); Banerjee et al. (2019) investigate how to achieve more accurate recognition on low-resolution or corrupted/noisy images, but did not resort to the use of recognition loss. Wang et al. (2019) propose a method to make denoised images more accurately segmented. Liu et al. (2019) introduced a theoretical framework for classification-distortion-perception tradeoff and conducted experiments with simulated or toy datasets, while our work develops practical approaches for real-world datasets. Most existing works only consider one image processing task or image domain, and develop specific techniques, while our simpler approach is task-agnostic and more widely applicable.

Our work is related but different from those aiming for robustness of the recognition model (Hendrycks & Dietterich, 2019; Li et al., 2019; Shankar et al., 2018), as we focus on training the processing models. Our method also shares some similarity with Palacio et al. (2018) which tries to differentiate input signals by optimizing recognition accuracy using auto-encoders. Bai et al. (2018); Zhang et al. (2018a); Liu et al. (2018); Sharma et al. (2018); Bai et al. (2018); Liu et al. (2017); Li et al. (2018) jointly train a processing model (e.g., dehazing, face reconstruction) together with a recognition model (e.g., face recognition) to achieve better image processing and/or recognition quality. Our problem setting is different from these works, in that we assume we do not have the control of the recognition model, as it might be on the cloud or to be decided in the future, thus we adapt the image processing model only. This provides transferability and also ensures the recognition of natural images is not harmed. To our knowledge, our work is the first to analyze such transferability in the literature. Appendix A includes our comparison with training recognition model jointly. We design and compare variants of our methods for different use cases, while prior studies only consider a single case. In addition, using extra discriminators have also been used in few-shot learning (Gidaris et al., 2019; Zhang et al., 2020) to enhance the feature quality and improve the performance on the original few-shot task, while our use of recognition model as loss is to improve image processing outputs' recognition accuracy, instead of the original processing task.

## 3 METHOD

We first introduce the problem setting of "recognition-aware" image processing, and then develop various approaches, each suited for different use cases. Our proposed methodology, although only introduced in a vision context, can be extended to other domains (e.g., speech) as well.

**Problem Setting.** In a typical image processing problem, given a set of training input images $\{I_{in}^k\}$ and corresponding target images $\{I_{target}^k\}$, we aim to train a network that maps an input to its target. Denoting this network as $P$ (processing), parameterized by $W_P$, our optimization objective is:

$$\min_{W_P} L_{proc} = \frac{1}{N} \sum_{k=1}^{N} l_{proc} \left( P\left(I_{in}^k\right), I_{target}^k \right), \tag{1}$$

where $l_{proc}$ is the loss function for each sample (e.g., $L_2$). The performance is typically evaluated by similarity (e.g., PSNR/SSIM) between $I_{target}^k$ and $P\left(I_{in}^k\right)$, or human assessment. In recognition-aware processing, we are interested in a recognition task, with a trained recognition model $R$ (recognition). We assume each target image $I_{target}^k$ is associated with a semantic label $S^k$ for the recognition task. Our goal is to train a processing model $P$ such that the recognition performance on the output images $P\left(I_{in}^k\right)$ is high, when evaluated using $R$ with the semantic labels $\{S^k\}$. In practice, $R$ might not be available (e.g., on the cloud), in which case we could resort to other models if the performance improvement transfers among models.

**Optimizing Recognition Loss.** Given our goal is to make the output images by $P$ more recognizable

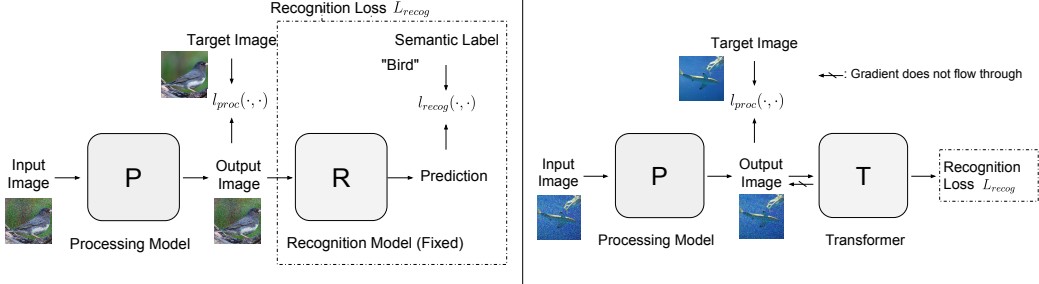

Figure 2: *Left*: RA (Recognition-Aware) processing. In addition to the image processing loss, we add a recognition loss using a fixed recognition model $R$, for the processing model $P$ to optimize. *Right*: RA with transformer. "Recognition Loss" stands for the dashed box in the left figure. A Transformer $T$ is introduced between the output of $P$ and input of $R$, to optimize recognition loss. We cut the gradient from recognition loss flowing to $P$, such that $P$ only optimizes the image processing loss and the image quality is not affected.

by $R$, it is natural to add a recognition loss on top of the objective of the image processing task (Eqn. 1) during training:

$$\min_{W_P} L_{recog} = \frac{1}{N} \sum_{k=1}^{N} l_{recog} \left( R \left( P \left( I_{in}^k \right) \right), S^k \right) \tag{2}$$

$l_{recog}$ is the per-example recognition loss defined by the downstream recognition task. For example, for image classification, $l_{recog}$ could be the cross-entropy (CE) loss. Adding the image processing loss (Eqn. 1) and recognition loss (Eqn. 2) together, our total training objective becomes

$$\min_{W_P} L_{proc} + \lambda L_{recog} \tag{3}$$

where $\lambda$ is the coefficient controlling the weights of $L_{recog}$ relative to $L_{proc}$. We denote this simple solution as "RA (Recognition-Aware) processing", which is visualized in Fig. 2 left. Note that once the training is finished, the recognition model used as loss is not needed anymore, and during inference, we only need the processing model P, thus no overhead is introduced in deployment. A potential shortcoming of directly optimizing $L_{recog}$ is that it might deviate $P$ from optimizing the original loss $L_{proc}$, and the trained $P$ will generate images that are not as good as if we only optimize $L_{proc}$. We will show that, however, with proper choice of $\lambda$, we could substantially boost the recognition performance with nearly no sacrifice on image quality.

If using $R$ as a fixed loss can only boost the accuracy on $R$ itself, the use of the method could be limited. Sometimes we do not have the knowledge about the future downstream recognition model or even task. Interestingly, we find that processing models trained with the loss of one recognition model $R_1$, can also boost the performance when evaluated using another model $R_2$, even if model $R_2$ has a different architecture, recognizes different categories or even performs a different task. This makes our method effective even when we cannot access the target downstream model, where we could use another trained model as the loss function. This phenomenon also implies that the "recognizability" of a processed image can be more general than just the extent it fits to a specific model.

**Unsupervised Optimization of Recognition Loss.** The solution above requires semantic labels for training images, which however, may not be always available. In this case, we could regress the recognition model's output of the target image $R(I_{target}^k)$, The recognition objective changes to

$$\min_{W_P} L_{recog} = \frac{1}{N} \sum_{k=1}^{N} l_{dis} \left( R \left( P \left( I_{in}^k \right) \right), R \left( I_{target}^k \right) \right) \tag{4}$$

where $l_{dis}$ is a distance measure between two of $R$'s outputs (e.g., $L_2$ distance, KL divergence). We call this approach "unsupervised RA". Note that it is only unsupervised for training model $P$, but not necessarily for the model $R$. The (pre)training of the model $R$ is not our concern since in our problem setting $R$ is a given trained model, and it can be trained either with or without full supervision. Unsupervised RA is related to the "knowledge distillation" paradigm (Hinton et al., 2014) used for network model compression, where the output of a large model is used to guide a small model, given the same input images. Instead we use the same recognition model $R$, but guide the upstream processing model to generate desirable images. It is also related to the perceptual loss/feature loss used in Johnson et al. (2016); Ledig et al. (2017), where the feature distance is minimized instead of output distance. We provide a comparison in Appendix A.

**Using an Intermediate Transformer.** Sometimes we want to prevent the added recognition loss $L_{recog}$ from causing $P$ to deviate from optimizing its original loss. We can achieve this by introducing an intermediate transformation model $T$: $P$'s output is first fed to the $T$, and $T$'s output serves as the input for $R$ (Fig. 2 right). $T$'s parameters $W_T$ are optimized for the recognition loss:

$$\min_{W_T} L_{recog} = \frac{1}{N} \sum_{k=1}^{N} l_{recog} \left( R \left( T \left( P \left( I_{in}^k \right) \right) \right), S^k \right) \tag{5}$$

With the help of $T$ on optimizing the recognition loss, the model $P$ can now "focus" on its original image processing loss $L_{proc}$. The optimization objective becomes:

$$\min_{W_P} L_{proc} + \min_{W_T} \lambda L_{recog} \tag{6}$$

In Eqn. 6, $P$ is solely optimizing $L_{proc}$ as in the original image processing problem (Eqn. 1). $P$ is learned as if there is no recognition loss, and therefore the image processing quality of its output will not be affected. This could be implemented by "detaching" the gradient generated by $L_{recog}$ between the model $T$ and $P$ (Fig. 2 right). We term this solution as "RA with transformer". Its downside compared with directly optimization using $P$ is that there are two instances for each image (the output of model $P$ and $T$), one is "for human" and the other is "for machines". Therefore, the transformer is best suited when we want to guarantee the image processing quality not affected at all, at the expense of maintaining another image. For example, in classifying images, we can have the higher-quality image presented to users for better experience and the other image passed to the backend for accurate machine classification.

## 4 EXPERIMENTS

We experiment on three image processing tasks: image super-resolution (SR), denoising, and JPEG-deblocking (17 tasks from ImageNet-C (Hendrycks & Dietterich, 2019) in Appendix F).

To obtain the input images, for SR, we downsample images by a factor of $4\times$; for denoising, we add Gaussian noise with 0.1 standard deviation; for JPEG deblocking, a quality factor of 10 is used for compression. We pair these three tasks with two visual recognition tasks, ImageNet classification and PASCAL VOC object detection. We adopt SRResNet (Ledig et al., 2017) as the architecture of the image processing model $P$ (more architectures in Appendix C). For the transformer model $T$, we use the ResNet-like architecture in CycleGAN (Zhu et al., 2017). The recognition models are ResNet, VGG and DenseNet. Please refer to Appendix A for comparison with alternative approaches and B for more details on training/hyperparameter settings.

**Evaluation on the Same Recognition Model.** We first present results when the $R$ used for evaluation is the same as the $R$ we use as the recognition loss in training. Table 1 shows our results on ImageNet classification. ImageNet-pretrained classification models ResNet-18/50/101, DenseNet-121 and VGG-16 are denoted as R18/50/101, D121, V16. "No Processing" denotes the accuracy on input images (low-resolution/noisy/JPEG-compressed); "Plain Processing" denotes using image processing models trained without recognition loss (Eqn. 1). We observe that plain processing can boost the accuracy over unprocessed images. These two are considered as baselines. Note there exist instances where the unprocessed images are classified correctly but the plain processing outputs are classified incorrectly. For example, on super-resolution with ResNet-18 as the recognition model, these instances account for 6.14% of the validation set. However, plain image processing does help the recognition accuracy on average.

Table 1: Accuracy (%) on ImageNet classification. R18 means ResNet-18, etc. The five models achieve 69.8, 76.2, 77.4, 74.7, 73.4 on original images. RA processing techniques substantially boost recognition accuracy.

| Task | Super-resolution | | | | | Denoising | | | | | JPEG-deblocking | | | | |
|---|---|---|---|---|---|---|---|---|---|---|---|---|---|---|---|
| Classification Model | R18 | R50 | R101 | D121 | V16 | R18 | R50 | R101 | D121 | V16 | R18 | R50 | R101 | D121 | V16 |
| No Processing | 46.3 | 50.4 | 55.5 | 51.6 | 42.1 | 46.8 | 55.8 | 61.3 | 59.7 | 46.7 | 43.1 | 47.7 | 55.2 | 49.2 | 43.9 |
| Plain Processing | 52.6 | 58.8 | 61.9 | 57.7 | 50.2 | 61.9 | 68.0 | 69.1 | 66.4 | 60.9 | 48.2 | 53.8 | 56.0 | 52.9 | 42.4 |
| RA Processing | 61.8 | 67.3 | 69.6 | 66.0 | 61.9 | 65.1 | 71.2 | 72.7 | 69.8 | 66.5 | 57.7 | 63.6 | 65.8 | 62.3 | 56.7 |
| RA, Unsupervised | 61.3 | 66.9 | 69.4 | 65.3 | 61.0 | 61.7 | 68.6 | 70.8 | 67.1 | 63.6 | 53.8 | 60.4 | 63.4 | 59.7 | 53.1 |
| RA w/ Transformer | 63.0 | 68.2 | 70.1 | 66.5 | 63.0 | 65.2 | 70.9 | 72.3 | 69.6 | 65.9 | 59.8 | 65.1 | 66.7 | 63.9 | 58.7 |

From Table 1, using RA processing can significantly boost the accuracy of output images over plainly processed ones, for all image processing tasks and recognition models. This is more prominent when the accuracy of plain processing is lower, e.g., in SR and JPEG-deblocking, where we mostly obtain ~10% accuracy gain (close to 20% in relative terms). Even without semantic labels, our unsupervised RA can still in most cases outperform baseline methods, despite achieves lower accuracy than its supervised counterpart. Also in SR and JPEG-deblocking, using an intermediate transformer $T$ can bring additional improvement over RA processing. See Appendix D for results on detection.

**Transfer between Recognition Architectures.** In reality, sometimes the $R$ we want to eventually evaluate the output images on might not be available for us to use as a loss for training, e.g., it could be on the cloud, kept confidential or decided later. In this case, we could train an processing model $P$ using recognition model $R_A$ (source) that is accessible to us, and after we obtain the trained model $P$, evaluate its output images' accuracy using another unseen $R_B$ (target). We evaluate model architecture pairs on ImageNet in Table 2, for RA Processing, where row is the source model ($R_A$), and column is the target model ($R_B$). In Table 2's each column, training with any model $R_A$ produces substantially higher accuracy than plainly processed images on $R_B$, indicating that the improvement is transferable among recognition architectures. This phenomenon enables us to use RA processing without the knowledge of the downstream recognition architecture.

Table 2: Transfer between recognition architectures. A processing model trained with source model $R_A$ (row) as recognition loss can improve the recognition performance on target model $R_B$ (column).

| Task | Super-resolution | | | | | Denoising | | | | | JPEG-deblocking | | | | |
|---|---|---|---|---|---|---|---|---|---|---|---|---|---|---|---|
| Train / Evaluation | R18 | R50 | R101 | D121 | V16 | R18 | R50 | R101 | D121 | V16 | R18 | R50 | R101 | D121 | V16 |
| Plain Processing | 52.6 | 58.8 | 61.9 | 57.7 | 50.2 | 61.9 | 68.0 | 69.1 | 66.4 | 60.9 | 48.2 | 53.8 | 56.0 | 52.9 | 42.4 |
| RA w/ R18 | 61.8 | 66.7 | 68.8 | 64.7 | 58.2 | 65.1 | 70.6 | 71.9 | 69.1 | 63.8 | 57.7 | 62.3 | 64.3 | 60.7 | 52.8 |
| RA w/ R50 | 59.3 | 67.3 | 68.8 | 64.3 | 59.1 | 64.2 | 71.2 | 72.2 | 69.2 | 64.7 | 55.8 | 63.6 | 64.7 | 61.0 | 53.5 |
| RA w/ R101 | 58.8 | 66.0 | 69.6 | 63.4 | 58.2 | 64.0 | 70.5 | 72.7 | 68.9 | 64.8 | 54.9 | 61.5 | 65.8 | 60.3 | 52.8 |
| RA w/ D121 | 59.0 | 65.6 | 67.8 | 66.0 | 57.4 | 64.2 | 70.6 | 72.0 | 69.8 | 64.3 | 54.8 | 61.8 | 64.4 | 62.3 | 52.9 |
| RA w/ V16 | 57.9 | 64.8 | 67.0 | 63.0 | 61.9 | 63.9 | 70.4 | 72.0 | 68.8 | 66.5 | 54.5 | 60.9 | 63.1 | 59.7 | 56.7 |

**Transfer between Object Categories.** What if the $R_A$ and $R_B$ recognize different categories of objects? We divide the 1000 classes from ImageNet into two splits, denoted as Cat (category) $A/B$, each with 500 classes, and train two 500-way classifiers (R18) on both splits, obtaining $R_A$ and $R_B$. Next, we train two image processing models $P_A/P_B$ with the $R_A/R_B$ as recognition loss, using images from Cat $A/B$ respectively. Note that neither $P$ nor $R$ has seen images/categories from the other split. We evaluate obtained processing models on both splits in Table 3. We observe that RA still benefits the accuracy even when transferring across categories (e.g., in SR, 60.1% to 66.5% transferring from Cat $A$ to Cat $B$). The improvement is only marginally lower than directly training on the same categories (e.g., 60.2% to 67.8% on Cat $B$). This suggests RA processing models do not impose category-specific signals to the images, but signals that enable wider sets of classes to be better recognized.

Table 3: Transfer between different object categories (500-way accuracy %). Refer to text for details.

| Task | Super-res | | Denoising | | JPEG-deblock | |
|---|---|---|---|---|---|---|
| Train/Eval | Cat $A$ | Cat $B$ | Cat $A$ | Cat $B$ | Cat $A$ | Cat $B$ |
| Cat $A$ Plain | 59.6 | 60.1 | 67.6 | 68.0 | 54.2 | 55.5 |
| Cat $A$ RA | 67.2 | 66.5 | 69.7 | 69.4 | 63.0 | 62.3 |
| Cat $B$ Plain | 59.6 | 60.2 | 67.0 | 67.5 | 54.7 | 56.0 |
| Cat $B$ RA | 66.4 | 67.8 | 69.4 | 69.7 | 62.1 | 63.5 |

Table 4: Transfer from ImageNet classification to PASCAL VOC object detection (mAP). Note that rows are classification models and columns are detection models, so even the same name in row and column (e.g., "R18") indicates different models trained on different tasks and datasets.

| Task | Super-resolution | | | | Denoising | | | | JPEG-deblocking | | | |
|---|---|---|---|---|---|---|---|---|---|---|---|---|
| Train / Evaluation | R18 | R50 | R101 | V16 | R18 | R50 | R101 | V16 | R18 | R50 | R101 | V16 |
| Plain Processing | 68.5 | 69.7 | 73.1 | 63.2 | 68.1 | 71.6 | 74.1 | 65.7 | 62.4 | 65.6 | 69.5 | 58.3 |
| RA w/ R18 | 71.3 | 73.5 | 75.6 | 67.8 | 70.6 | 73.1 | 75.5 | 64.1 | 67.7 | 70.3 | 73.2 | 62.4 |
| RA w/ R50 | 70.8 | 73.2 | 74.8 | 67.8 | 70.4 | 73.1 | 75.8 | 66.2 | 67.8 | 70.2 | 73.1 | 62.8 |
| RA w/ R101 | 70.7 | 73.2 | 75.3 | 67.0 | 70.5 | 73.5 | 75.7 | 66.9 | 68.1 | 70.2 | 72.8 | 63.2 |
| RA w/ D121 | 71.2 | 73.6 | 75.3 | 67.2 | 70.5 | 73.2 | 75.7 | 65.7 | 68.1 | 70.5 | 73.1 | 62.6 |
| RA w/ V16 | 70.4 | 72.4 | 74.6 | 67.5 | 70.6 | 73.0 | 75.7 | 67.7 | 67.8 | 70.3 | 73.2 | 63.7 |

**Transfer between Recognition Tasks and Datasets.** We evaluate task transferability when task $A$ is classification and task $B$ is object detection in Table 4, where rows are classification models used for RA loss and columns are detection models for evaluation. There is also a dataset shift, since model $P$ and $R$ are both trained on ImageNet; during evaluation, $P$ is fed with VOC images and we use a VOC-trained detection model $R$. We observe that using classification loss on model $A$ (row) gives accuracy gain on model $B$ over plain processing in most cases. Such task transferability suggests the "machine semantics" of the image could be a task-agnostic property.

**Transfer to an Black-box, Third-party Cloud Model.** We compare the images generated from plain processing and RA models using the "General" model at `clarifai.com`, a company providing state-of-the-art image classification cloud services. We do not have knowledge of the model's architecture or what datasets it was trained on, except we can access the service using APIs. The model also recognizes over 11000+ concepts that are different from the 1000-class ImageNet categories. For this experiment, we only take the output category with the maximum probability as the prediction. We use the SR processing model trained with R18/ImageNet as the RA model. We randomly sample image indices from ImageNet validation set, and ask clarifai.com for predictions of both images generated from plain and RA processing models.

From the results, we then randomly select 100 instances where clarifai.com gives different predictions on plain and RA images, to compose a survey for user study. For each of the 100 instances, the survey presents the user with the target image, and both prediction labels generated from plain/RA images, in randomized left/right order. The survey asks the user to indicate in his/her opinion which label(s) describe the image to a satisfactory level. The user has the options to choose none, either or both labels. The survey and instructions can be found at `https://tinyurl.com/y698779q`. 10 volunteers participated in our survey. The resulting average satisfaction rates for plain and RA super-resolved images are 40.1% and 55.3% respectively. We achieve 15.2% absolute gain or 37.9% relative gain on recognition satisfaction rate, indicating the strong transferability our method provides without knowledge of the black-box cloud model.

## 5  ANALYSIS AND CONCLUSION

**Image Processing Quality Assessment.** We compare the output image quality using conventional metrics (PSNR/SSIM). When using RA with transformer, the output quality of $P$ is guaranteed unaffected, therefore here we evaluate RA processing. We use R18 as loss on ImageNet, and report results with different $\lambda$s (Eqn. 3) in Table 5. $\lambda = 0$ corresponds to plain processing. When

Table 5: PSNR/SSIM/Accuracy using different $\lambda$s.

| $\lambda$ | Super-resolution | Denoising | JPEG-deblocking |
|---|---|---|---|
| 0 | **26.73/0.805**/52.6 | **31.24/0.895**/61.9 | **27.50/0.825**/48.2 |
| $10^{-4}$ | 26.69/0.804/59.2 | 31.18/0.894/64.4 | **27.50**/0.823/56.0 |
| $10^{-3}$ | 26.31/0.792/**61.8** | 30.78/0.884/**65.1** | 27.17/0.810/**57.7** |
| $10^{-2}$ | 25.47/0.760/61.3 | 29.71/0.855/64.3 | 26.32/0.776/56.6 |

$\lambda = 10^{-4}$, PSNR/SSIM are only marginally worse. However, the accuracy obtained is significantly higher. This suggests that the added recognition loss is not harmful when $\lambda$ is chosen properly. When $\lambda$ is excessively large ($10^{-2}$), image quality is hurt more, and interestingly even the recognition accuracy start to decrease, which could be due to the change of actual learning rate. A proper balance between processing and recognition loss is needed for both image quality and accuracy. We also measure the image quality using the PieAPP metric (Prashnani et al., 2018), which emphasizes more on perceptual difference: on SR, when $\lambda = 0/10^{-4}/10^{-3}$, PieAPP (lower is better) = 1.329/1.313/1.323. Interestingly, RA processing can slightly *improve* perceptual quality measured with PieAPP.

In Fig. 3, we visualize some examples where the output image is incorrectly classified with a plain processing model, but correctly recognized with RA processing. With smaller $\lambda$ ($10^{-2}$ and $10^{-3}$), the image is nearly the same as the plainly processed images. When $\lambda$ is too large ($10^{-2}$), we could see some extra textures when zooming in. More results are presented in Appendix H.

**Decision Boundaries Analysis and Transferability.** Inspired by prior works' analysis on adversarial example transferability (Liu et al., 2016; Tramèr et al., 2017), we conduct *decision boundary* analysis to gain insights on RA processing's transferability. The task used is SR with ImageNet. We restrict our analysis to a single direction at a time due to image's high dimension: given a input image $x$ and a direction $d$ (unit vector, same dimension as $x$), we analyze how the output of the recognition model $R$ changes when $x$ moves along $d$ by $\delta$ amount, i.e., when input is $x + \delta s$.

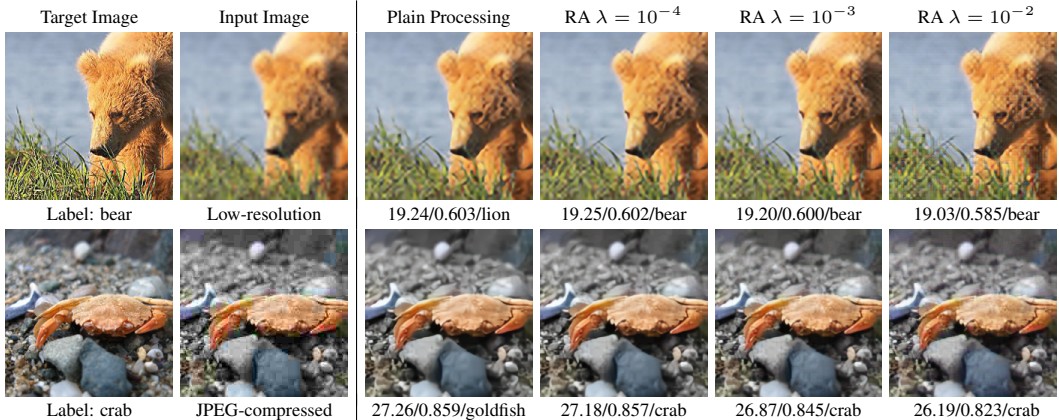

Figure 3: Examples where outputs of RA processing models can be correctly classified but those from plain processing models cannot. PSNR/SSIM/class prediction is shown below each output image. Slight differences between images from plain processing and RA processing models could be noticed when zoomed in.

We define the *boundary distance* (BD) of model $R$, with respect to input $x$ and direction $s$, as the minimum amount of movement along $d$ required for $x$ to produce a different label at $R$, or more formally: $\text{BD}(R, x, d) = \arg\min_{\delta > 0}\{R(x + \delta \cdot d) \neq R(x)\}$.

Consider a two-model scenario, with a source model $R_s$ and a target model $R_t$ sharing the same output categories. We define their *inter-boundary distance* (IBD): $\text{IBD}(R_s, R_t, x, d) = |\text{BD}(R_s, x, d) - \text{BD}(R_t, x, d)|$. Intuitively, if $R_s(x) = R_t(x)$ (same prediction within boundary), a small IBD between $R_s$ and $R_t$ means they have close boundaries along the $s$ direction, since $x$ does not need to move beyond one's boundary too far to reach the other's. In this case, changes made to $x$ along $d$ likely has a *transferring* effect from source to target model due to their close boundaries.

We take the image $x$ to be a plain processing output, and consider two types of directions: 1. random direction $d_r$. 2. The direction generated by subtracting the plain processing output $x$ from RA processing output $x_s$, i.e., $(x_s - x)/||x_s - x||_2$. The RA processing model here is trained with the source model $R_s$, thus $x_s$ is specific to $R_s$. We call this "RA direction" ($d_{RA}$), since it points to the RA output $x_s$ from the plain output $x$. We take all validation images such that the plain processing output $x$ generates the same wrong prediction when fed to $R_s$ and $R_t$, i.e., $R_s(x) = R_t(x) \neq$ Ground Truth. For each image, we compute $\text{BD}(R_s, x, d), \text{BD}(R_t, x, d)$ and $\text{IBD}(R_s, R_t, x, d)$ with $d$ being random direction and RA direction. In this experiment we present results with $R_s$ being R18 and $R_t$ being R50, as we observe other model pairs produce similar trends.

Table 6: Decision boundary analysis results.

| Direction | Random | RA |
|---|---|---|
| $\text{BD}(R_s)$ | 34.7 | 14.1 |
| $\text{BD}(R_t)$ | 34.3 | 16.5 |
| $\text{IBD}(R_s, R_t)$ | 25.2 | 10.5 |

The average results are shown in Table 6. We first observe that BDs are much smaller alongside the RA direction than the random direction. This indicates moving along the RA direction will change the model's wrong prediction at $x$ faster, possibly to a correct prediction. More importantly, under either random or RA direction, IBD is always smaller than source/target BDs, which indicates $R_s$ and $R_t$'s boundaries are relatively close, leading to a transfering effect. This result in RA direction can explain why RA processing can lead to transferable accuracy gains, since the RA loss brings this direction as the effect on the processing output $x$.

We further visualize decision boundaries in Fig. 4 with examples (see Appendix I for more). We use R18 as source and each of the other models as target. Here we plot $d_{RA}$ as the horizontal and $d_r$ as the vertical axis. The origin represents the plain processing output $x$, and the color of point $(u, v)$ represents the predicted class of the image $x + u \cdot d_{RA} + v \cdot d_r$. From the plot, we can see that different models share similar decision boundaries, and also tend to change to the same prediction once we move from the origin along a direction far enough. In both examples, we do confirm that when we move towards RA direction (towards right at horizontal axis), the first color we encounter (green for top, purple for bottom) represents the image's correct label. This suggests the signal from RA loss (RA direction) can correct the wrong prediction with plain processing output ($x$ at origin), and such correction is transferable given the similar decision boundaries among models.

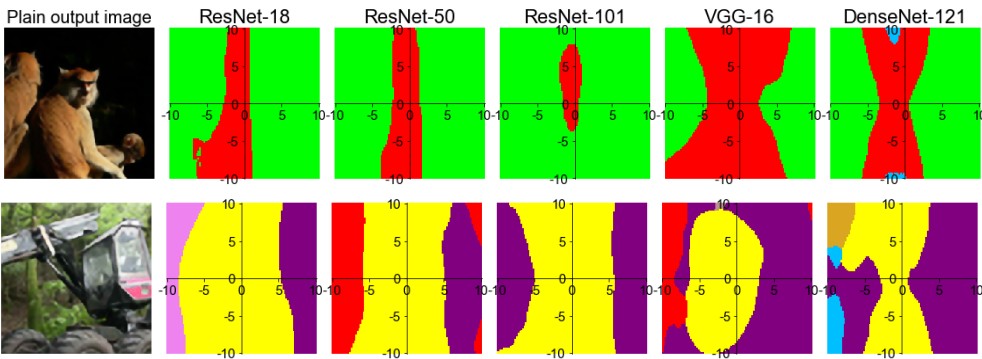

Figure 4: Different models' decision boundaries are similar, especially along the RA direction (horizontal axis).

**Conclusion.** We investigated an important yet largely ignored problem: enhancing the machine interpretability of image processing outputs. We found that our simple approaches – optimizing with the additional recognition loss, can significantly boost the recognition accuracy with minimal or no loss in image quality. Moreover, such gain can transfer to architectures, object categories, vision tasks unseen during training, or even a black-box cloud model, indicating the enhanced interpretability is not specific to one particular model but generalizable to others. This makes the approaches applicable even when the future downstream recognition models are unknown. Finally we analyzed the decision boundary similarities of recognition models to explain this transferability phenomenon.

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

# APPENDIX

## TABLE OF CONTENTS

## A  COMPARISON WITH ALTERNATIVES

We analyze some alternatives to our approaches. Unless otherwise specified, experiments in this section are conducted using RA processing on super-resolution, with ResNet-18 trained on ImageNet as the recognition model, and $\lambda = 10^{-3}$ if used. Under this setting, we achieve 61.8% classification accuracy on the output images.

**Training/Fine-tuning the Recognition Model.** Instead of fixing the recognition model $R$, we could train/fine-tune it together with the training of image processing model $P$, to optimize the recognition loss. Many prior works (Sharma et al., 2018; Bai et al., 2018; Zhang et al., 2018a) do train/fine-tune the recognition model jointly with the image processing model. We use SGD with momentum as $R$'s optimizer, and the final accuracy reaches 63.0%. However, since we do not fix $R$, it becomes a model that specifically recognizes super-resolved images, and we found its performance on original target images drops from 69.8% to 60.5%. Moreover, when transferring the trained $P$ on ResNet-50, the accuracy is 62.4 %, worse than 66.7% when we train with a fixed ResNet-18. This suggests we lose some transferability if we do not fix the recognition model $R$.

**Training Recognition Models from Scratch.** We could first train a super-resolution model, and then train $R$ from scratch on the output images. Doing this, we achieve 66.1% accuracy, higher than 61.8% in RA processing. However, $R$'s accuracy on original clean images drops from 69.8% to 66.1%. Alternatively, we could train $R$ from scratch on interpolated low-resolution images, in which case we achieve 66.0% on interpolated validation data but only 50.2% on the original data. In summary, training/fine-tuning $R$ to cater the need of super-resolved or interpolated images can harm its performance on original images, and causes additional overhead in storing models. In contrast, RA processing could boost the accuracy of output images with the performance on original images intact.

**Training without the Image Processing Loss.** It is possible to train the processing model on the recognition loss $L_{recog}$, without even keeping the original image processing loss $L_{proc}$ (Eqn. 3). This may presumably lead to better recognition performance since the model $P$ can now "focus on" optimizing the recognition loss. However, we found removing the original image processing

loss hurts the recognition performance: the accuracy drops from 61.8% to 60.9%; even worse, the SSIM/PSNR metrics drop from 26.69/0.804 to 16.92/0.263, which is reasonable since the image processing loss is not optimized during training. This suggests the original image processing loss is helpful for the recognition accuracy, since it helps the corrupted image to restore to its original form.

**Perceptual/Feature Loss.** Our unsupervised RA method optimizes the recognition model's output probability distance between processed and target images. This is related to the perceptual loss (also called feature loss) used in Johnson et al. (2016); Ledig et al. (2017). Perceptual loss optimizes processed and target images' distance in VGG feature space. Note that the perceptual loss was originally proposed to improve output's quality from a human observer's perspective. To compare both methods, we follow Ledig et al. (2017) to optimize the perceptual loss from VGG-16. We find perceptual loss yields lower accuracy than unsupervised RA (56.7% vs. 61.0% on the VGG-16 recognition model). This could be because using final probabilities provides more semantic supervision, while intermediate features improve the outputs from a perceptual perspective.

# B EXPERIMENTAL DETAILS

**General Setup.** We evaluate our proposed methods on three image processing tasks: image super-resolution, denoising, and JPEG-deblocking. In those tasks, the target images are all the original images from the datasets. To obtain the input images, for super-resolution, we use a downsampling scale factor of $4\times$; for denoising, we add Gaussian noise on the images with a standard deviation of 0.1 to obtain the noisy images; for JPEG deblocking, a quality factor of 10 is used to compress the image to JPEG format. The image processing loss used is the mean squared error (MSE, or $L_2$) loss. For the recognition tasks, we consider image classification and object detection, two common tasks in computer vision. In total, we have 6 ($3 \times 2$) task pairs to evaluate. Training is performed with the training set and results on the validation set are reported.

We adopt the SRResNet (Ledig et al., 2017) as the architecture of the image processing model $P$, which is simple yet effective in optimizing the MSE loss. Even though SRResNet is originally designed for super-resolution, we find it also performs well on denoising and JPEG deblocking when its upscale parameter set to 1 for the same input-output sizes. Throughout the experiments, on both the image processing network and the transformer, we use the Adam optimizer (Kingma & Ba, 2014) with an initial learning rate of $10^{-4}$, following the original SRResNet (Ledig et al., 2017). Our implementation is in PyTorch (Paszke et al., 2017). Each result is based on one run due to the relatively stable performance (almost all within 1%) we noticed and the large amount of tasks combinations/architectures to be evaluated. For example, on super-resolution with ResNet-18 as $R$ on ImageNet, five runs on plain processing gives accuracies (%): 53.9, 53.8, 54.0, 53.8, 53.9 (53.88±0.07); RA processing: 61.7, 61.9, 61.6, 61.8, 61.7 (61.74±0.10); unsupervised RA: 61.2, 61.3, 61.3, 61.4, 61.1 (61.26±0.10); RA w/ Transformer: 62.9, 63.0, 62.9, 62.9, 63.0 (62.94±0.05).

The experiments are run on 1-4 NVIDIA TITAN Xp GPUs. Clearly training time is linear in the iterations trained and the space taken is a constant with a fixed input size. The training process finishes in 2-24 hours depending on the model sizes/variants of methods/recognition tasks, and the maximum GPU memory taken is 30GB (multi-GPU) with batch size of 20.

**Image Classification.** For image classification, we evaluate our method on the large-scale ImageNet benchmark (Deng et al., 2009), which can be downloaded at http://image-net.org/download. It consists of $\sim 1.2$ million training images and 50,000 validation images. We use five pre-trained image classification models, ResNet-18/50/101 (He et al., 2016), DenseNet-121 (Huang et al., 2017) and VGG-16 (Simonyan & Zisserman, 2015) with BN (Ioffe & Szegedy, 2015) (denoted as R18/50/101, D121, V16 in Table 1), on which the top-1 accuracy (%) of the original validation images is 69.8, 76.2, 77.4, 74.7, and 73.4 respectively. We train the processing models for 6 epochs on the training set, with a learning rate decay of $10\times$ at epoch 5 and 6, and a batch size of 20. In evaluation, we feed unprocessed validation images to the image processing model, and report the accuracy of the output images evaluated on the pre-trained classification networks. For unsupervised RA, we use $L_2$ distance as the function $l_{dis}$ in Eqn. 4. The hyperparameter $\lambda$ is chosen using super-resolution with the ResNet-18 recognition model, on two small subsets for training/validation from the original large training set, from a grid search from $10^{-4}$ to 100. The $\lambda$ chosen for RA processing, RA with transformer, and unsupervised RA is $10^{-3}$, $10^{-2}$ and 10 respectively.

**Object Detection.** For object detection, we evaluate on PASCAL VOC 2007 and 2012 detection dataset (https://pjreddie.com/projects/pascal-voc-dataset-mirror/), using Faster-RCNN (Ren et al., 2015) as the recognition model. Our implementation is based on the code from (Yang et al., 2017). Following common practice (Redmon et al., 2016; Ren et al., 2015; Dai et al., 2016), we use VOC 07 and 12 trainval data as the training set, and evaluate on VOC 07 test data. The Faster-RCNN training uses the same hyperparameters in (Yang et al., 2017). For the recognition model's backbone architecture, we evaluate ResNet-18/50/101 and VGG-16 (without BN (Ioffe & Szegedy, 2015)), obtaining mAP of 74.2, 76.8, 77.9, 72.2 on the test set respectively. Given those trained detectors as recognition loss functions, we train the models on the training set for 7 epochs, with a learning rate decay of $10 \times$ at epoch 6 and 7, and a batch size of 1. We report the mean Average Precision (mAP) of processed images in the test set. As in image classification, we use $\lambda = 10^{-3}$ for RA processing, and $\lambda = 10^{-2}$ for RA with transformer.

## C  RESULTS ON MORE ARCHITECTURES

In our main paper, we use SRResNet (Ledig et al., 2017) as our processing model $P$. Here we provide more results with other more recent processing models, including SRDenseNet (SRDNet) (Tong et al., 2017), Residual Dense Network (RDN) (Zhang et al., 2018d), and Deep Back-Projection Networks (DBPN) Haris et al. (2018). We present results at Table 7, with super-resolution as the processing task, ImageNet classification as recognition task, and $R$ being ResNet-18.

Table 7: Accuracy (%) on ResNet-18 ImageNet classification, with more processing architectures. Processing task is supre-resolution.

| $P$ Architecture | SRResNet | SRDNet | RDN | DBPN |
|---|---|---|---|---|
| No Processing | 46.3 | 46.3 | 46.3 | 46.3 |
| Plain Processing | 52.6 | 54.7 | 55.5 | 54.8 |
| RA Processing | 61.8 | 62.4 | 63.8 | 62.6 |
| RA, Unsupervised | 61.3 | 61.7 | 63.6 | 62.4 |
| RA w/ Transformer | 63.0 | 63.2 | 64.1 | 63.5 |

## D  RESULTS ON OBJECT DETECTION (SAME ARCHITECTURE)

The results for object detection, when evaluated on the same architecture, are shown in Table 8. We observe similar trend as in classification: using recognition loss can consistently improve the mAP over plain image processing by a notable margin. On super-resolution, RA processing mostly performs on par with RA with transformer, but on the other two tasks using a transformer is slightly better. The model with transformer performs better more often possibly because with this extra network in the middle, the capacity of the whole system is increased.

Table 8: mAP on VOC object detection. The four models achieve 74.2, 76.8, 77.9, 72.2 on original images.

| Task | Super-resolution | | | | Denoising | | | | JPEG-deblocking | | | |
|---|---|---|---|---|---|---|---|---|---|---|---|---|
| Detection Model | R18 | R50 | R101 | V16 | R18 | R50 | R101 | V16 | R18 | R50 | R101 | V16 |
| No Processing | 67.9 | 70.3 | 72.1 | 63.6 | 51.8 | 56.5 | 61.8 | 38.9 | 49.3 | 54.5 | 64.1 | 38.4 |
| Plain Processing | 69.2 | 70.7 | 73.3 | 64.2 | 68.9 | 72.0 | 74.7 | 65.8 | 63.7 | 66.5 | 70.4 | 60.3 |
| RA Processing | 71.2 | 74.4 | 75.6 | 68.1 | 70.9 | 73.7 | 75.6 | 67.6 | 67.4 | 70.4 | 72.9 | 63.9 |
| RA w/ Transformer | 71.4 | 74.2 | 75.6 | 66.0 | 71.0 | 73.9 | 75.9 | 67.7 | 68.5 | 70.7 | 73.7 | 64.4 |

## E  MORE RESULTS ON TRANSFERABILITY

We present additional results on transferability in this section.

Table 9: Transfer between recognition architectures, evaluated on PASCAL VOC object detection (mAP).

| Task | Super-resolution | | | | Denoising | | | | JPEG-deblocking | | | |
|---|---|---|---|---|---|---|---|---|---|---|---|---|
| Evaluation on | R18 | R50 | R101 | V16 | R18 | R50 | R101 | V16 | R18 | R50 | R101 | V16 |
| Plain Processing | 69.2 | 70.7 | 73.3 | 64.2 | 68.9 | 72.0 | 74.7 | 65.8 | 63.7 | 66.5 | 70.4 | 60.3 |
| RA w/ R18 | 71.2 | 73.8 | 75.2 | 66.9 | 70.9 | 74.0 | 75.5 | 67.2 | 67.4 | 70.0 | 72.3 | 63.5 |
| RA w/ R50 | 70.6 | 74.4 | 75.4 | 66.4 | 70.6 | 73.7 | 75.5 | 67.2 | 67.0 | 70.4 | 72.4 | 63.2 |
| RA w/ R101 | 71.1 | 73.8 | 75.6 | 65.8 | 70.3 | 73.6 | 75.6 | 66.2 | 65.9 | 69.3 | 72.9 | 61.3 |
| RA w/ V16 | 70.4 | 72.8 | 74.9 | 68.1 | 69.9 | 73.4 | 75.6 | 67.6 | 66.1 | 69.3 | 72.1 | 63.9 |

### E.1 TRANSFERRING BETWEEN ARCHITECTURES

We provide the model transferability results of RA processing on object detection in Table 9. Rows indicate the models trained as recognition loss and columns indicate the evaluation models. We see similar trend as in classification (Table 1): using other architectures as loss can also improve recognition performance over plain processing; the loss model that achieves the highest performance is mostly the model itself, as can be seen from the fact that most boldface numbers are on the diagonals.

In Table 10, we present the results when transferring between recognition architectures, using unsupervised RA. We note that for super-resolution and JPEG-deblocking, similar trend holds as in (supervised) RA processing, as using any architecture in training will improve over plain processing. But for denoising, this is not always the case. Some models $P$ trained with unsupervised RA are slightly worse than the plain processing counterpart. A possible reason for this is the noise level in our experiments is not large enough and plain processing achieve very high accuracy already.

Table 10: Transfer between recognition architectures using unsupervised RA, on ImageNet classification.

| Task | Super-resolution | | | | | Denoising | | | | | JPEG-deblocking | | | | |
|---|---|---|---|---|---|---|---|---|---|---|---|---|---|---|---|
| Evaluation on | R18 | R50 | R101 | D121 | V16 | R18 | R50 | R101 | D121 | V16 | R18 | R50 | R101 | D121 | V16 |
| Plain Processing | 52.6 | 58.8 | 61.9 | 57.7 | 50.2 | 61.9 | 68.0 | 69.1 | 66.4 | 60.9 | 48.2 | 53.8 | 56.0 | 52.9 | 42.4 |
| Unsup. RA w/ R18 | 61.3 | 66.3 | 68.6 | 64.5 | 57.3 | 61.7 | 67.9 | 69.7 | 66.4 | 60.5 | 53.8 | 59.1 | 62.0 | 57.5 | 50.0 |
| Unsup. RA w/ R50 | 58.9 | 66.9 | 68.6 | 64.1 | 58.2 | 61.2 | 68.6 | 70.3 | 66.6 | 61.3 | 52.8 | 60.4 | 62.5 | 58.3 | 50.3 |
| Unsup. RA w/ R101 | 57.8 | 64.9 | 69.0 | 62.9 | 56.9 | 60.6 | 68.0 | 70.7 | 66.3 | 60.7 | 52.3 | 58.7 | 63.4 | 57.9 | 49.0 |
| Unsup. RA w/ D121 | 58.0 | 64.7 | 67.2 | 65.3 | 56.0 | 60.7 | 67.8 | 69.7 | 67.1 | 60.3 | 52.2 | 59.2 | 62.2 | 59.7 | 49.9 |
| Unsup. RA w/ V16 | 57.7 | 64.6 | 67.3 | 63.2 | 61.0 | 60.4 | 67.1 | 69.6 | 65.9 | 63.6 | 52.0 | 58.4 | 61.5 | 57.4 | 53.1 |

In Table 11, we present the results of transferring between architectures when we use a transformer $T$. We use the processing model $P$ and transformer $T$ trained with $R_A$ together when evaluating on $R_B$. From Table 11, in most cases improvement is still transferable but there are a few exceptions. For example, when $R_A$ is ResNet or DenseNet and when $R_B$ is VGG-16, in most cases the accuracy fall behind plain processing by a large margin. This weaker transferability is possibly caused by the fact that there is no constraint imposed by the image processing loss on $T$'s output, thus it "overfits" more to the specific $R$ it is trained with.

Table 11: Transfer between architectures using RA with Transformer ($T$), on ImageNet classification.

| Task | Super-resolution | | | | | Denoising | | | | | JPEG-deblocking | | | | |
|---|---|---|---|---|---|---|---|---|---|---|---|---|---|---|---|
| Evaluation on | R18 | R50 | R101 | D121 | V16 | R18 | R50 | R101 | D121 | V16 | R18 | R50 | R101 | D121 | V16 |
| Plain Processing | 52.6 | 58.8 | 61.9 | 57.7 | 50.2 | 61.9 | 68.0 | 69.1 | 66.4 | 60.9 | 48.2 | 53.8 | 56.0 | 52.9 | 42.4 |
| RA w/$T$ w/ R18 | 63.0 | 59.2 | 67.0 | 63.9 | 27.0 | 65.2 | 69.4 | 71.6 | 68.4 | 40.3 | 59.8 | 58.7 | 62.6 | 60.3 | 19.9 |
| RA w/$T$ w/ R50 | 60.5 | 68.2 | 68.9 | 65.8 | 40.4 | 63.1 | 70.9 | 71.5 | 68.6 | 48.7 | 55.0 | 65.1 | 63.9 | 61.9 | 31.5 |
| RA w/$T$ w/ R101 | 59.6 | 66.2 | 70.1 | 65.1 | 35.6 | 62.4 | 68.8 | 72.3 | 67.6 | 52.3 | 54.8 | 61.3 | 66.7 | 24.8 | 60.5 |
| RA w/$T$ w/ D121 | 58.5 | 64.2 | 66.9 | 66.5 | 27.3 | 58.0 | 66.8 | 67.3 | 69.6 | 46.7 | 46.6 | 57.2 | 59.0 | 63.9 | 9.0 |
| RA w/$T$ w/ V16 | 59.2 | 64.7 | 67.8 | 65.0 | 63.0 | 57.6 | 64.0 | 67.1 | 55.7 | 63.1 | 56.1 | 61.2 | 63.4 | 58.7 | 60.1 |

### E.2 TRANSFERRING BETWEEN RECOGNITION TASKS

In Section 4, we investigated the transferability of improvement from classification to detection. Here we evaluate the opposite direction, from detection to classification. The results are shown in Table 12. Here, using RA processing can still consistently improve over plain processing for any pair of models, but we note that the improvement is not as significant as directly training using classification models as loss (Table 1 and Table 2).

Table 12: Transfer from PASCAL VOC object detection to ImageNet classification (accuracy %). A image processing model $P$ trained with detection model $A$ (row) as recognition loss can improve the performance on classification model $B$ (column) over plain processing.

| Task | Super-resolution | | | | | Denoising | | | | | JPEG-deblocking | | | | |
|---|---|---|---|---|---|---|---|---|---|---|---|---|---|---|---|
| Evaluation on | R18 | R50 | R101 | D121 | V16 | R18 | R50 | R101 | D121 | V16 | R18 | R50 | R101 | D121 | V16 |
| Plain Processing | 53.0 | 58.9 | 62.0 | 57.3 | 50.9 | 59.7 | 65.1 | 67.3 | 63.9 | 59.2 | 48.8 | 54.6 | 56.8 | 53.1 | 44.7 |
| RA w/ R18 | 54.6 | 60.2 | 63.4 | 58.8 | 52.7 | 60.8 | 66.7 | 68.8 | 65.2 | 61.1 | 50.8 | 57.2 | 59.6 | 55.4 | 48.5 |
| RA w/ R50 | 54.0 | 59.7 | 63.0 | 58.7 | 52.0 | 60.5 | 66.6 | 68.5 | 64.9 | 60.8 | 50.7 | 56.9 | 59.2 | 55.3 | 48.3 |
| RA w/ R101 | 54.1 | 59.8 | 63.3 | 58.7 | 52.5 | 60.2 | 66.1 | 68.3 | 64.6 | 60.6 | 51.3 | 57.2 | 59.5 | 55.5 | 48.3 |
| RA w/ V16 | 54.5 | 60.4 | 63.6 | 59.1 | 52.7 | 60.4 | 66.6 | 68.4 | 64.7 | 60.6 | 50.6 | 56.5 | 58.7 | 54.9 | 47.9 |

Additionally, the results when we transfer the model $P$ trained with unsupervised RA with image classification to object detection are shown in Table 13. In most cases, it improves over plain processing, but for image denoising, this is not always the case. Similar to results in Table 10, this could be because the noise level is relatively low in our experiments.

Table 13: Transfer from ImageNet classification to PASCAL VOC object detection, using unsupervised RA.

| | Super-resolution | | | | Denoising | | | | JPEG-deblocking | | | |
|---|---|---|---|---|---|---|---|---|---|---|---|---|
| Evaluation on | R18 | R50 | R101 | V16 | R18 | R50 | R101 | V16 | R18 | R50 | R101 | V16 |
| Plain Processing | 68.5 | 69.7 | 73.1 | 63.2 | 68.1 | 71.6 | 74.1 | 65.7 | 62.4 | 65.6 | 69.5 | 58.3 |
| Unsup. RA w/ R18 | 71.3 | 73.4 | 75.3 | 66.8 | 69.0 | 71.3 | 74.3 | 61.1 | 65.2 | 68.1 | 71.3 | 59.8 |
| Unsup. RA w/ R50 | 70.7 | 73.3 | 75.0 | 66.6 | 68.9 | 71.7 | 74.4 | 63.1 | 65.4 | 68.5 | 71.2 | 60.0 |
| Unsup. RA w/ R101 | 70.7 | 73.2 | 75.0 | 66.2 | 68.9 | 71.3 | 73.9 | 63.3 | 65.2 | 67.9 | 71.1 | 59.6 |
| Unsup. RA w/ D121 | 71.0 | 73.2 | 75.1 | 66.6 | 68.7 | 70.3 | 73.0 | 63.8 | 65.9 | 68.6 | 71.4 | 61.1 |
| Unsup. RA w/ V16 | 70.3 | 72.3 | 74.3 | 67.0 | 68.5 | 70.7 | 74.0 | 63.6 | 65.9 | 68.2 | 71.5 | 61.1 |

## F RESULTS ON IMAGENET-C

We evaluate our methods on the ImageNet-C benchmark (Hendrycks & Dietterich, 2019). It imposes 17 different types of corruptions on the ImageNet (Deng et al., 2009) validation set. Despite ImageNet-C benchmark is designed for more robust recognition models, but not for testing image processing models, it is a good testbed to test our methods in a broader range of processing tasks. Since only corrupted images from the validation set are released, we divide it evenly for each class into two halves and train/test on its first/second half. The corrupted image is the input image to the processing model and the original clean image is the target image. The recognition model used in this experiment is an ImageNet-pretrained ResNet-18.

Table 14: ImageNet-C results (top-1 accuracy %) under different types of corruptions with corruption level 5.

| Type | orig | brit | contr | defoc | elast | gau_b | gau_n | glass | impul | jpeg | motn | pixel | shot | satr | snow | spat | speck | zoom |
|---|---|---|---|---|---|---|---|---|---|---|---|---|---|---|---|---|---|---|
| No Processing | 69.9 | 51.3 | 3.3 | 11.3 | 17.1 | 9.3 | 1.2 | 8.7 | 1.0 | 29.4 | 11.1 | 23.1 | 1.8 | 39.5 | 10.7 | 19.1 | 7.7 | 17.6 |
| Plain Processing | N/A | 59.9 | 18.3 | 25.3 | 18.9 | 21.5 | 21.8 | 20.1 | 24.1 | 43.0 | 42.4 | 50.1 | 24.9 | 54.4 | 34.5 | 60.8 | 36.6 | 17.0 |
| RA Processing | N/A | 61.4 | 30.7 | 33.8 | 35.4 | 27.0 | 32.8 | 25.3 | 35.1 | 46.1 | 48.2 | 54.0 | 35.2 | 57.1 | 43.7 | 63.0 | 45.2 | 31.9 |

In Table 14, we evaluate RA Processing on all 17 types of corruptions, with corruption level 5 as in (Hendrycks & Dietterich, 2019). We observe that RA Processing brings consistent improvement over plain processing, sometimes by an even larger margin than the tasks considered in Sec. 4.

Table 15: ImageNet-C results (top-1 accuracy %) under different levels of corruptions, with corruption level "snow" and "speckle noise".

| Corruption Type | Snow | | | | | Speckle noise | | | | |
|---|---|---|---|---|---|---|---|---|---|---|
| Corruption Level | 1 | 2 | 3 | 4 | 5 | 1 | 2 | 3 | 4 | 5 |
| No Processing | 46.7 | 23.6 | 28.0 | 17.6 | 10.7 | 50.5 | 42.8 | 22.9 | 14.5 | 7.7 |
| Plain Processing | 57.1 | 45.1 | 46.0 | 37.1 | 34.5 | 60.3 | 57.0 | 48.4 | 43.2 | 36.6 |
| RA Processing | 60.3 | 51.7 | 51.7 | 45.7 | 43.7 | 62.7 | 60.8 | 54.2 | 50.3 | 45.2 |
| Unsupervised RA | 60.2 | 51.3 | 50.6 | 43.6 | 41.5 | 62.9 | 60.5 | 53.8 | 49.4 | 43.9 |
| RA w/ Transformer | 55.7 | 46.7 | 48.1 | 42.7 | 40.9 | 59.0 | 57.7 | 52.2 | 49.2 | 44.7 |

In Table 15, we experiment with different levels of corruptions with corruption type "speckle noise" and "snow". We also evaluate with our variants – Unsupervised RA and RA with Transformer. We observe that when the corruption level is higher, our methods tend to bring more recognition accuracy gain. In this case, we note that using a Transformer could sometimes hurt the accuracy compared with plain processing. This is possibly because the insufficient training data in ImageNet-C dataset caused the transformer to hurt, since more parameters typically require more training data. In the majority of other cases, it improves slightly over RA processing.

Table 16: ImageNet-C results (top-1 accuracy %) under different levels of corruptions, with corruption level "snow" and "speckle noise".

| Corruption Type | Snow | | | | | Speckle noise | | | | |
|---|---|---|---|---|---|---|---|---|---|---|
| Evaluation on | R18 | R50 | R101 | D121 | V16 | R18 | R50 | R101 | D121 | V16 |
| No Processing | 10.7 | 16.6 | 20.9 | 21.7 | 10.5 | 7.7 | 11.7 | 14.5 | 18.6 | 7.1 |
| Plain Processing | 34.5 | 39.1 | 44.6 | 41.1 | 27.4 | 36.6 | 42.4 | 47.7 | 43.0 | 31.3 |
| RA w/ R18 | 43.7 | 47.9 | 51.7 | 47.9 | 37.4 | 45.2 | 50.3 | 53.3 | 49.1 | 39.0 |

In Table 16, we examine the transferability of RA Processing between recognition architectures, using the same two tasks "speckle noise" and "snow", with corruption level 5. Note the recognition loss used during training is from a ResNet-18, and we evaluate the improvement over plain processing on ResNet-50/101, DenseNet-121 and VGG-16. We observe that the improvement over plain processing is transferable among different architectures.

## G  EXPERIMENTS ON SEMANTIC SEGMENTATION

We experiment with the Cityscapes (Cordts et al., 2016) semantic segmentation task using the recent HRNet-W48 Wang et al. (2020) architecture as the recognition model. As with other tasks, we use a $\lambda$ of 0.001 for RA processing, and SRResNet as processing model. We train for 18 epochs and other settings are the same as in the main paper.

| Processing Model | Plain | RA |
|---|---|---|
| mIoU | 66.18 | 71.10 |
| PSNR | 33.24 | 33.31 |
| SSIM | 0.9159 | 0.9154 |

Table 17: RA Processing on Cityscapes semantic segmentation.

Table 17 compares the recognition accuary (mIoU) of Plain and RA models. We observe that RA processing is able to improve the acuracy substantially, with similar image qualities in terms of PSNR/SSIM.

| Processing Model | Plain | RA |
|---|---|---|
| mIoU | 72.28 | 76.66 |
| PSNR | 35.43 | 35.22 |
| SSIM | 0.9403 | 0.9326 |

Table 18: Transferring processing models from ImageNet classification to Cityscapes semantic segmentation.

We also evaluate the transferring effect from ImageNet classification to Cityscapes semantic segmentation. We take a plain processing model and an RA processing model (ResNet-18 as recognition model) trained with ImageNet classification, and compare their outputs' performance on Cityscapes segmentation. Table 18 shows, using RA Processing on ImageNet classification can help segmentation accuracy on Cityscapes as well. We also note both accuracies are higher than when we train the processing model on Cityscapes. This is possibly due to the abundance of data in ImageNet (1.2 million images) compared with Cityscapes (3k images).

## H MORE IMAGE QUALITY VISUALIZATIONS

We provide more visualizations in Fig. 5 where the output image is incorrectly classified by ResNet-18 with a plain image processing model, and correctly recognized with RA processing, as in Fig. 3 at Section 5.

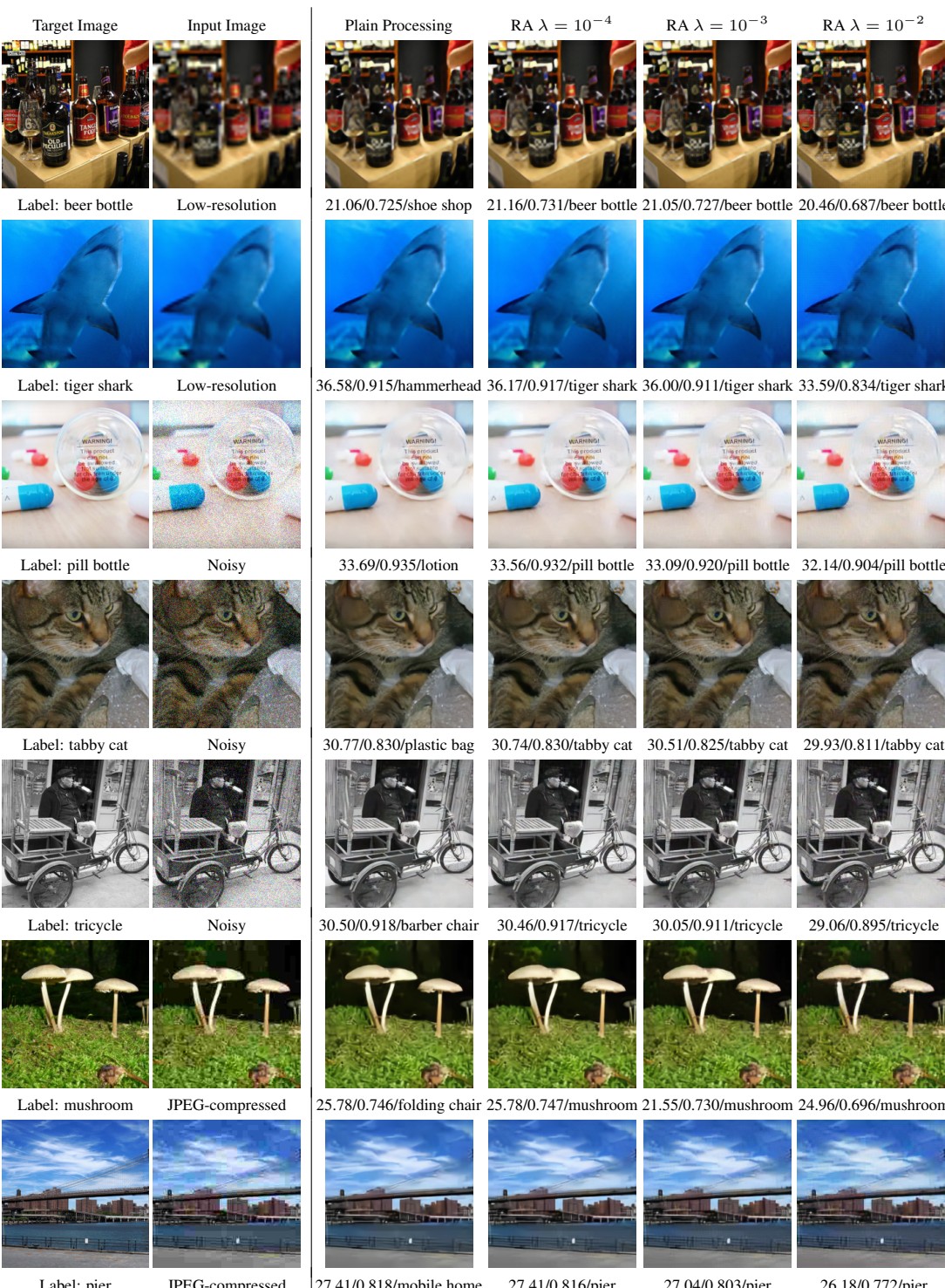

Figure 5: Examples where output images from RA processing models can be correctly classified but those from plain processing models cannot. PSNR/SSIM/class prediction is shown below each output image. Slight differences between images from plain processing and RA processing models (especially with large $\lambda$s) could be noticed when zoomed in.

# I   MORE DECISION BOUNDARY VISUALIZATIONS

We provide more visualizations in Fig. 6 of different model's decision boundaries to complement Fig. 4. We can see that different recognition models share similar boundaries.

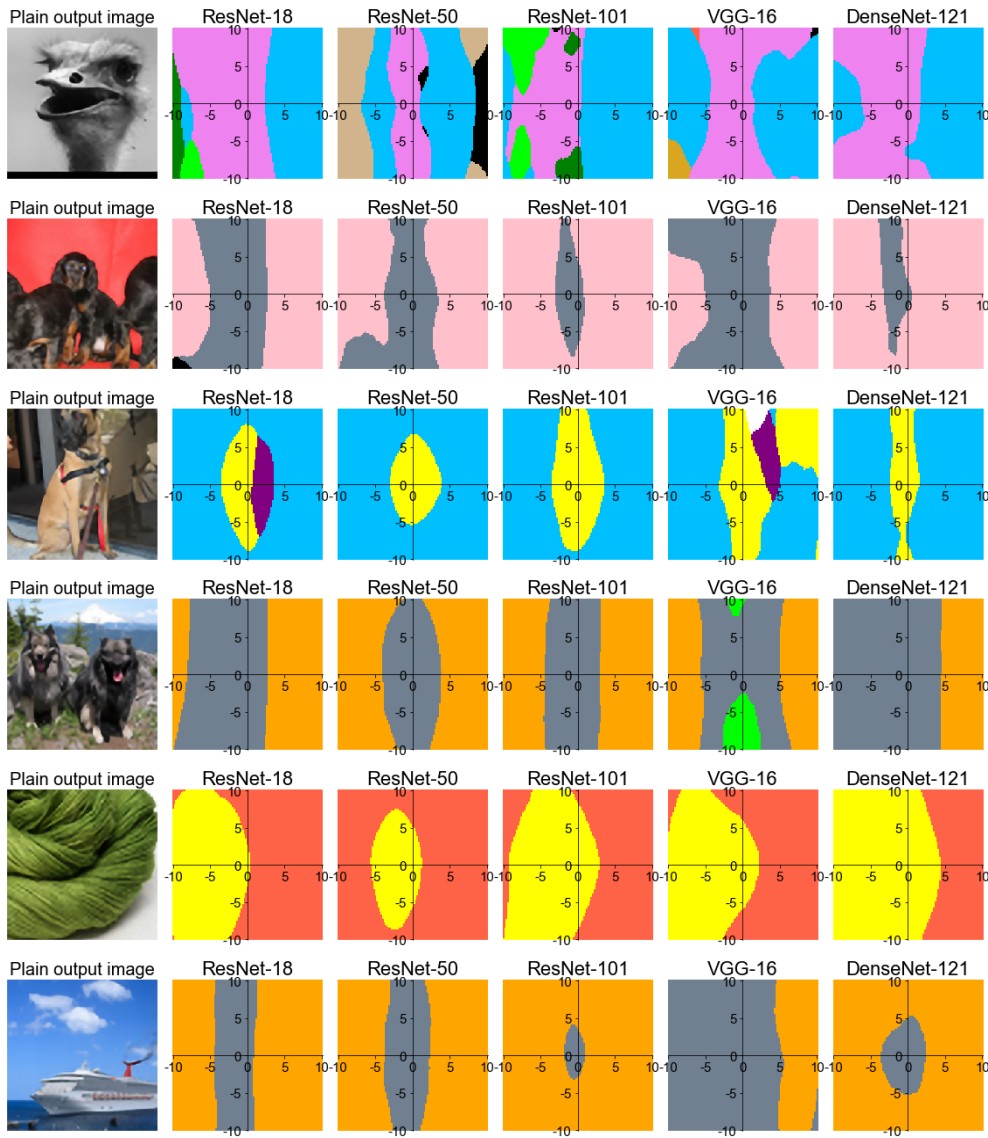

Figure 6: Different models' decision boundaries are similar, especially along the RA direction (horizontal axis).

