# OpenReview forum: "Transferable Recognition-Aware Image Processing"
_ICLR.cc/2021/Conference — Reject_

### Official Review · AnonReviewer4 · 2020-10-29
**Interesting Paper**

**Rating:** 6
**Confidence:** 3

**Review:**

The paper proposed a learnable image processing methods that improve machine interpretability of processed image. The paper mainly claimed that improvement of machine recognition is transferrable when evaluated on models of different architectures, recognized categories, tasks and training datasets. Additionally, the paper also try to explain this transferability phenomenon by demonstrating the similarities of different models’ decision boundaries.

Pros:
(1) The writing is clear and easy to follow. The problem is well formulated and experiments is comparably extensive.
(2) The paper proposed a unsupervised RA scheme without the Recognition label, which maximize the agreement of recognition output between original and processed image, this approach sounds novel to me.
(3) The Decision Boundaries Analysis and Transferability part is interesting, to my knowledge is the first work that to investigate the transferability of RA.

Cons:
(1) The problem that high-level task can benefits low-level task has already been well-studied in multiple literature [1][2][3][4], the first half of the paper is just a revisit of those methods while swaping differen backbones and tasks. I acknowledge that 3 representative low-level tasks has been studied, however for high-level part the paper only studied recognition and detection task, more semantically intensive task such as segmentation is missing.
(2) The paper simply adding a transformer in between Process module and Recogition module, to guarantee the image processing quality not affected. This seem to be a trivial approach and need more experiment to justify.


[1] Sreya Banerjee, Rosaura G VidalMata, Zhangyang Wang, and Walter J Scheirer. Report on ugˆ 2+ challenge track 1: Assessing algorithms to improve video object detection and classiﬁcation from unconstrained mobility platforms. arXiv preprint arXiv:1907.11529, 2019. 3
[2] Sicheng Wang, Bihan Wen, Junru Wu, Dacheng Tao, and Zhangyang Wang. Segmentation-aware image denoising without knowing true segmentation. arXiv preprint arXiv:1905.08965, 2019.
[3] Boyi Li, Xiulian Peng, Zhangyang Wang, Jizheng Xu, and Dan Feng. End-to-end united video dehazing and detection. In Thirty-Second AAAI Conference on Artiﬁcial Intelligence, 2018.
[4] Ding Liu, Bihan Wen, Xianming Liu, Zhangyang Wang, and Thomas S Huang. When image denoising meets high-level vision tasks: A deep learning approach. arXiv preprint arXiv:1706.04284, 2017.

---

> ### Author Response · Authors · 2020-11-18
> **Thank you for the review; addressing concerns below**
>
> We thank you for your valuable feedback! We are glad you find our problem well formulated, experiments extensive, method (Unsupervised RA) novel, and decision boundary analysis interesting. We reply point-by-point below and we will follow up by posting a revision of the paper during the discussion period.
>
> &nbsp;
>
> > (1) The problem that high-level task can benefits low-level task has already been well-studied in multiple literature [1][2][3][4], the first half of the paper is just a revisit of those methods while swapping different backbones and tasks. I acknowledge that 3 representative low-level tasks has been studied, however for high-level part the paper only studied recognition and detection task, more semantically intensive task such as segmentation is missing.
>
> **First we would like to note that the pointed references [1,2,3,4] are cited and discussed in the related works (Section 2).** We acknowledge that there are prior studies on the interaction between image processing and recognition, but we would like to highlight some differences between our study and the pointed prior works [1,2,3,4]:
>
> 1. **Our work directly optimizes recognition accuracy with recognition loss while most entry solutions in [1] focus on developing better restoration and enhancement techniques to indirectly optimize recognition performance,** without resorting the use of downstream recognition models as losses.
>
> 2. **We use a fixed pre-trained recognition model as additional loss function for image processing and show that the accuracy gain can transfer, while [3,4] use a “joint training” approach for the processing and recognition models and did not study transferability.** As we noted in related work, this is an important distinction since we assume we do not have the knowledge of the recognition model, as it might be on the cloud or to be decided in the future, thus we adapt the image processing model only. Also, joint training of the recognition model **hurts its accuracy on natural images, and the accuracy gain’s transferability,** as we showed in Appendix A “Training/Fine-tuning the Recognition Model”.
>
> 3. **We design and compare variants of our methods (RA processing, unsupervised, transformer) for different use cases,** while prior studies only consider a single case.
>
> 4. **We demonstrated our approaches’ wide applicability through a variety of task pairs while [2,3,4] each focuses on a specific processing and recognition task pair**(e.g., denoising + segmentation in [2], dehazing + detection in [3]).
>
> 5. **Our work also suggests that accuracy gain transfers among recognition tasks,** potentially reducing the need to train RA processing models for each specific recognition task, while [2,3,4] does not consider this issue and only demonstrates the improvement on one particular task.
>
> We will highlight these differences more clearly in the revision.
>
> Due to space limitations we did not include more recognition tasks other than image classification and object detection. We are currently experimenting with semantic segmentation on Cityscapes dataset, and we will post the result/update the submission (Appendix) when the results are ready.
>
> &nbsp;
>
> > (2) The paper simply adding a transformer in between Process module and Recogition module, to guarantee the image processing quality not affected. This seem to be a trivial approach and need more experiment to justify.
>
> **First we would like to note the Transformer approach is only one variant of our approaches and not the whole scope of our work,** as we developed multiple variants for different use cases, including RA processing, unsupervised RA and the Transformer variant. We also extensively investigated the transferability phenomenon in multiple scenarios and analyzed potential reasons from decision boundary similarities’ perspective. These results do not simply follow from the design of the Transformer approach.
>
> **We believe our study on the Transformer approach is quite extensive, with experiments on ImageNet classification (Table 1), PASCAL VOC detection (Table 8), ImageNet-C classification (Table 15)  and its transferability properties (Table 11).** We also view the simplicity of the Transformer model as an advantage as it is easier to deploy in practice and achieves our goal of guaranteeing image processing quality. If there are further experiments to justify its effectiveness we will be happy to include them.
>
> Thank you again for the review and we hope our response can address your concerns.

---

> > ### Author Response · Authors · 2020-11-25
> > **Update on semantic segmentation experiments**
> >
> > **We have experimented with the task of semantic segmentation in the discussion period.** We use the recent HRNet-W48 model [1][2] as the recognition architecture, and the same SRResNet in our submission as the super resolution model. The segmentation dataset used is Cityscapes. As with other tasks, we use a $\lambda$ of 0.001 for RA processing. We train for 18 epochs and other settings are the same as in the submission.
> >
> > The table below compares the recognition accuracy (mIoU) of Plain and RA models. We observe that RA processing is able to improve the accuracy substantially, with similar image qualities in terms of PSNR/SSIM.
> >
> >
> > | Processing Model | Plain| RA  |
> > |:-:|:------:|:------:|
> > | **mIOU**    | 66.18 | 71.10 |
> > | **PSNR**    | 33.24 | 33.31 |
> > | **SSIM**    | 0.9159 | 0.9154 |
> >
> >
> >
> > **We also evaluate the transferring effect from ImageNet classification to Cityscapes semantic segmentation.** We take a plain processing model and an RA processing model (ResNet-18 as recognition model) trained with ImageNet classification, and compare their outputs' performance on Cityscapes segmentation. The table below shows, using RA Processing on ImageNet classification can help segmentation accuracy on Cityscapes as well. We also note both accuracies are higher than when we train the processing model on Cityscapes. This is possibly due to the abundance of data in ImageNet (1.2m images) compared with Cityscapes (3k images).
> >
> >
> > | Processing Model | Plain| RA  |
> > |:-:|:------:|:------:|
> > | **mIOU**    | 72.28 | 76.66 |
> > | **PSNR**    | 35.43 | 35.22 |
> > | **SSIM**    | 0.9403 | 0.9326 |
> >
> >
> > In the updated submission, we have included these initial results in Appendix G. We will include more results in our next version.
> >
> > [1] Deep High-Resolution Representation Learning for Visual Recognition.
> > [2] https://github.com/HRNet/HRNet-Semantic-Segmentation

---

### Official Review · AnonReviewer1 · 2020-10-30
**An interesting paper but the novelty is somewhat not enough**

**Rating:** 5
**Confidence:** 4

**Review:**

This paper proposes an interesting image processing approach to deal with the issue that the processed images cannot be recognized by machines. To address this issue,  the author proposes the so-called recognition-aware image processing strategy in which an extra discriminator is employed to classify the semantic annotation of given samples.

Strength:
-  The idea is technically sound. Such a simple design can theoretically improve the quality of processed images in either supervised or unsupervised manner.
-  The paper is easy to understand and follow.
-  The experiments are solid to demonstrate the usefulness of the proposed method. The authors evaluate their design on three different image processing scenarios, i. e., super-resolution, denoising, deblocking, and the improvements in all scenarios are impressive.

Weakness:
- The novelty is somewhat limited. The idea proposed in this paper is very similar to the motivation of multi-task learning and also the ideas in several few-shot learning works [1,2], which employ extra discriminators to improve the feature quality. To this end, the contribution of this paper maybe not enough.
- Is there any clear illustration of why an image processing network has negative impacts on recognition accuracy?

[1] Boosting Few-Shot Visual Learning with Self-Supervision

[2] Rethinking Class Relations: Absolute-relative Few-shot Learning

---

> ### Author Response · Authors · 2020-11-18
> **Thank you for the review; addressing concerns below**
>
> We thank you for your valuable feedback! We are glad you find the idea technically sound, experiments solid, and improvements impressive. We reply point-by-point below and we will follow up by posting a revision of the paper during the discussion period.
>
> &nbsp;
>
> >1. The novelty is somewhat limited. The idea proposed in this paper is very similar to the motivation of multi-task learning and also the ideas in several few-shot learning works [1,2], which employ extra discriminators to improve the feature quality. To this end, the contribution of this paper maybe not enough.
>
> **Our contribution in this work includes the development of multiple approaches in different use cases, as well as pointing out the opportunity to use the recognition loss to improve image processing outputs’ recognition accuracy,** a largely overlooked problem in image processing. **We extensively investigated the transferability phenomenon with these approaches in multiple scenarios,** including across architectures, categories, tasks, and to a black-box model (Section 4). **We analyzed potential reasons for transferability from decision boundary similarities**(Section 5). **As pointed out by R4, the study on transferability is the first in the literature to our knowledge, and is a major part of our contribution.**
>
> We acknowledge that using multiple loss functions (as in multi-task learning) is not a new proposal in our submission. Using multiple losses can date back to very early stages of machine learning, and is prevalent in deep learning. However, **we believe the use of recognition loss in this specific problem to achieve transferable accuracy improvement on processed images is novel and not studied before.**
>
> Our methods share with the few-shot learning works [1,2] in that they use additional discriminators/losses, but our purpose is to improve image processing outputs’ recognition accuracy (not to enhance the image processing quality, the "original" task), while their purpose is to enhance the feature quality and boost the performance in the original task (few-shot learning). **Such use of extra discriminators is prevalent in deep learning,** for example in image classification [3] and semantic segmentation [4]. We also discussed and compared with a more related “perceptual loss” [5] in Section 2/Appendix A. Therefore, **we believe these methods in few-shot learning do not necessarily impact our novelty since we tackle a different problem for a different purpose.** We will add discussions to the few-shot learning works and highlight the differences in our revision.
>
> [3] Deeply-Supervised Nets. Lee et al.
> [4] Pyramid Scene Parsing Network. Zhao et al.
> [5] Perceptual Losses for Real-Time Style Transfer and Super-Resolution. Johnson et al.
>
> &nbsp;
>
> >2. Is there any clear illustration of why an image processing network has negative impacts on recognition accuracy?
>
> Indeed, **there exist instances where the unprocessed images are classified correctly but the plain processing outputs are classified incorrectly.** For example, on super-resolution with ResNet-18 as the recognition model, these instances account for 6.14% of the validation set. However, **plain image processing does help the recognition accuracy on average**: as shown in Table 1, the row “Plain Processing” generally outperforms the row “No Processing”. To clarify, our main point is **plain processing alone does not necessarily optimally help recognition accuracy, and our RA processing techniques can help further improve outputs’ accuracy.**
>
> Thank you again for the review and we hope our response can address your concerns.

---

### Official Review · AnonReviewer6 · 2020-11-06
**An interesting paper but not strong enough**

**Rating:** 5
**Confidence:** 4

**Review:**

This paper proposes a setting called "recognition-aware image processing." The key idea is to make the images output by image processing methods still be readily recognized by image recognition methods. Realizing this will help to better meet the requirement from both human observers and machines. Formally, this is formulated as a combined optimization in which the losses from image processing and recognition tasks are jointly considered. This framework is further extended to unsupervised case and the case of intermediate transformer to make it more flexible. The transferability issue is discussed and it is observed that the model trained by the proposed method can generally help even when other recognition models or tasks are used. Experimental study is conducted to demonstrate the performance of the proposed method.

The issue considered in this paper is interesting, and the proposed method is a good attempt to address this issue. The result shows that it is possible to balance between the performance of image processing and that of image recognition. This paper is well written and easy to follow.

Meanwhile,

1. This paper may need more solid cases to justify the necessity of such a setting in practice. The benefit of combining image processing and image recognition from this perspective needs to be better explained. As shown by the experiment, image processing quality is more or less affected, and a careful tuning of the tradeoff value may be needed. In this case, separating these two tasks may lead to an overall easier solution.

2. The technical contribution of this work is a bit limited. The essence of the proposed work is just a weighted combination of the image processing loss and image recognition loss, with some extension to the unsupervised case and the intermediate transformer case. However, the overall novelty does not seem to be sufficient.

3. The transferability highlighted in this work needs to be more rigorously analyzed and justified.

--- Thank the authors for the detailed response. This paper investigates an interesting issue. However, the overall technical contribution and novelty of this work remains to be a concern. After reading the response and the comments of peer reviewers, the rating is maintained as follows.

---

> ### Author Response · Authors · 2020-11-18
> **Thank you for the review; addressing concerns below**
>
> We thank you for the valuable feedback! We are glad you find the problem interesting and the methods are good attempts. We reply point-by-point below and will follow up by posting a revision of the paper during the discussion period.
>
> > 1. [...] need more solid cases to justify the necessity of such a setting in practice. The benefit of combining image processing and image recognition [...] needs to be better explained. [...] image processing quality is more or less affected, and a careful tuning of the tradeoff value may be needed. [...] separating these two tasks may lead to an overall easier solution.
>
> **Justification for the setting:**(1) **The immediate benefit of combining image processing and recognition is the substantial boost in recognition accuracy, and the gains are transferable**(Sec. 4, Table 1,2,3,4). Keeping the two tasks separate by only training recognition models on natural images loses such opportunity. (2) **If we separate these two tasks by specifically training a recognition model on processed images, the performance on natural images can be harmed**(Sec. 1, Para. 3). The experimental evidence of this can be found in Appendix A “Training Recognition Models from Scratch”. This “retraining the recognition model” scheme might also be impractical considering the significant overhead induced by catering to various image processing tasks and models, making it necessary to maintain multiple recognition models. (3) **It is not always feasible to separate two tasks by training a recognition model for processed images, because the recognition model may not be known for now, or not in our control**(Sec.1, Para. 5). For example, when the processed image is uploaded to the Internet, if it cannot be accurately recognized by models on the servers, it may be improperly handled by vision-based search engines, recommender systems or image taggers.
>
> **Image quality**: (1) **Under the PieAPP metric [1] which emphasizes more on perceptual quality than pixel distances between the images, RA processing can actually improve the performance**(Sec. 5). (2) **In terms of traditional metrics (PSNR/SSIM), the image quality is only hurt minorly**with a proper hyperparameter $\lambda$. Our methods only introduce one additional hyperparameter $\lambda$, and we believe a proper choice of hyperparameters is needed in most research. Because we demonstrate the transferability of improvement gain across various scenarios, there is also less need to tune $\lambda$ for each setting. Instead we could use the same $\lambda$ in many cases.
>
>
> > 2. The technical contribution of this work is a bit limited. The essence of the proposed work is just a weighted combination of the image processing loss and image recognition loss, with some extension to the unsupervised case and the intermediate transformer case. However, the overall novelty does not seem to be sufficient.
>
> **Our contribution includes the development of approaches in different use cases, as well as pointing out the opportunity to use the recognition loss to improve image processing outputs’ recognition accuracy,** a largely overlooked problem in image processing. **We extensively investigated the transferability phenomenon in multiple scenarios,** including across architectures, categories, tasks, and to a black-box model (Section 4). **We analyzed potential reasons for transferability from decision boundary similarities**(Section 5).
>
> We acknowledge that the methods are relatively straightforward, but **with their effectiveness we would like to view the this as an advantage for potential wider usage in practice,** and its usage as baselines in future studies, as we noted in introduction. We do not view the weighted combination of different losses as our main contribution, as it is prevalent in the deep learning community. However, **we believe the usage of recognition loss in this specific problem and the transferability it brings is novel and not thoroughly studied before.**
>
> > 3. The transferability highlighted in this work needs to be more rigorously analyzed and justified.
>
> **We believe that our analysis on transferability is extensive and in depth both in terms of the transferring scenarios (architectures, categories, tasks/datasets, black-box models, in Sec. 4), and the analysis on decision boundaries for the explanation.** For the latter, we provided both **quantitative**(distance measures, Table 6) and **qualitative**(boundary visualization, Figure 4) analysis for the evidence of the similarity in decision boundaries.**As R4 pointed out, our analysis on transferability is the first time in the literature, to our knowledge.**Similar analysis has been done in [2][3] to analyze the transferability of adversarial examples. If there is more that we can include, suggestions are welcomed!
>
> Thank you again for your review and we hope our response can address your concerns.

---

> > ### Author Response · Authors · 2020-11-25
> > **references**
> >
> > [1] PieAPP: Perceptual Image-Error Assessment through Pairwise Preference. Prashnani et al.
> > [2] Delving into Transferable Adversarial Examples and Black-box Attacks. Liu et al.
> > [3] The Space of Transferable Adversarial Examples. Tramèr et al.

---

### Author Response · Authors · 2020-11-25
**Summary of revision**

Thanks to all reviewers for their constructive comments! We have updated our submission accordingly. Here is a summary of the revisions:

1. We clarified our contributions and novelty (especially about transferability) more clearly in Section 1, in response to R1 and R6.
2. We highlighted the differences from related works pointed by R1 and R4 in Section 2.
3. We clarified on the justification of our setting and why the processing and recognition tasks may not be separated in Section 1, in response to R6's concern.
4. We added a note about the improvement of "plain processing" over "no processing" in Section 4 in response to R1's question.
5. We included experiments on semantic segmentation (Appendix G) in response to R4's comment.

We hope our responses and revisions can help address the concerns.

---

### Decision · Program_Chairs · 2021-01-07
**Final Decision**

**Decision:**

Reject

**Comment:**

Three reviewers have reviewed this paper and they maintain their findings after the rebuttal. The reviewers are mainly concerned about the novelty (several highly-related papers exist) and well as the technical contribution (more theoretical developments are needed). Therefore, this paper in its current form cannot be accepted.